# FINE-GRAINED TEXT-TO-IMAGE SYNTHESIS WITH SEMANTIC REFINEMENT

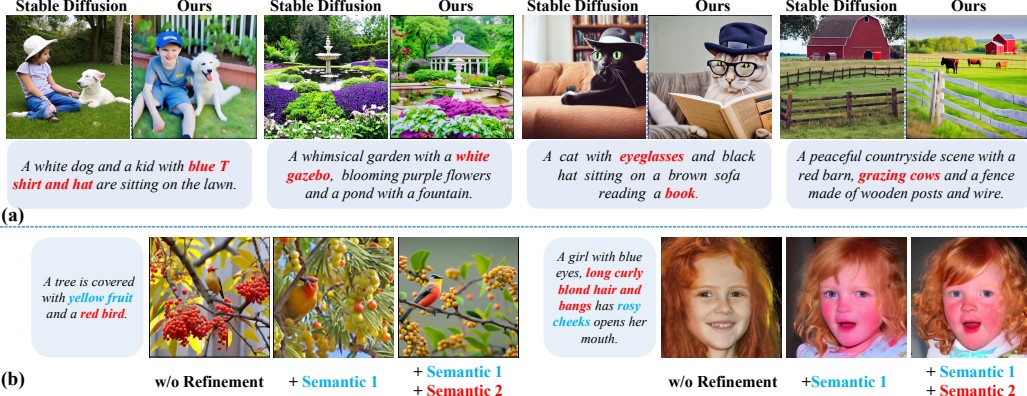

Figure 1: (a) Comparison between stable diffusion (Rombach et al., 2022) and our method, where stable diffusion fails to meet some detailed requirements (*e.g.*, highlighted text) given lengthy text prompt. (b) Fine-grained text-conditioned image synthesis with *semantic refinement*, where users are allowed to regularize the generation with some semantic details of their interests. Note that our approach supports emphasizing an arbitrary number of semantics *at one time* instead of performing refinement step by step. Thus, for each triplet in (b), the last two columns are *not* refined from their previous one, instead, all samples are produced independently.

## ABSTRACT

Recent advance in text-to-image synthesis greatly benefits from large-scale vision-language models such as CLIP. Despite the capability of producing high-quality and creative images, existing methods often struggle in capturing details of the text prompt, especially when the text is lengthy. We reveal that such an issue is partially caused by the imperfect text-image matching using CLIP, where fine-grained semantics may get obscured by the dominant ones. This work presents a new diffusion-based method that favors fine-grained synthesis with **semantic refinement**. Concretely, instead of getting a synthesis using the entire descriptive sentence as the prompt, users can emphasize some specific words of their own interests. For this purpose, we incorporate a semantic-induced gradient as a reference input in each denoising step to help the model understand the selected sub-concept. We find out that our framework supports the combination of multiple semantics by directly adding up their corresponding gradients. Extensive results on various datasets suggest that our approach outperforms existing text-to-image generation methods by synthesizing semantic details with finer granularity.

## 1 INTRODUCTION

The advent of large-scale vision-language models (Radford et al., 2021), together with the success of diffusion models (Ho et al., 2020; Song et al., 2020; Lu et al., 2022; Nichol & Dhariwal, 2021; Watson et al., 2021), facilitates the development of text-to-image synthesis (Ramesh et al., 2021; Zhou et al., 2022; Nichol et al., 2021; Ramesh et al., 2022b; Rombach et al., 2022; Sun et al., 2022). These models enable the generation of diverse, high-quality images that match textual prompts, even incorporating fantastic artistic styles (Hertz et al., 2022; Ruiz et al., 2022; Couairon et al., 2022; Gal et al., 2022; Chen et al., 2022; Kawar et al., 2022). Despite the remarkable achievement, existing

methods have been found to perform poorly when the description has rich details (as shown in Figure 1a), which appears as an incomplete match with the description.

To look into the cause of this issue, we revisit the working mechanism of vision-language models by taking the popular CLIP (Radford et al., 2021) model as an example. We find that, given a text, CLIP may *not* always assign the best matching image with the highest score, especially when the text is long. For example, although CLIP successfully pairs $S_1$ and $I_1$ in Figure 2, $S_2$ is mistakenly paired to $I_2$, which is neither "with beard" nor "serious". But at the same time, $I_3$ has high responses to both "serious" and "beard". This phenomenon implies that, from the viewpoint of an imperfectly learned vision-language model, the detailed information within a text may get concealed by some other key concepts in the description, such as "man" in $S_2$, making the text-image matching score unreliable.

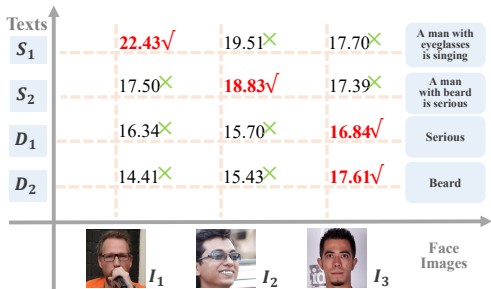

Figure 2: **Matching scores** between texts and images predicted by CLIP. Image $I_3$ fails to be paired with sentence $S_2$, but responds highly to the details (*i.e.*, $D_1$ and $D_2$) within $S_2$.

Inspired by the above analysis, we come up with a new diffusion-based generative model to facilitate fine-grained text-to-image synthesis. Our motivation is intuitive: since the one-time matching with CLIP scores may fail to capture every detail within the text condition, we aim to highlight some semantics (*e.g.*, words and phrases) to help guide the generation more accurately. We call such a process as *semantic refinement*, which is illustrated in Figure 1b. For this purpose, we redesign the denoising pipeline of diffusion models. Concretely, in each denoising step, the network takes a semantic-induced gradient as a reference input, alongside the image denoised from the previous step, to refine the generation from the semantic perspective. In this way, users can easily rectify the coarse-grained synthesis, which is predicted based on the *entire sentence*, with the gradients obtained from their *selected words*.

We evaluate our approach, termed as *SeReDiff*, on a range of datasets. Compared to existing alternatives, our approach is capable of producing fine-grained synthesis that better meets the input text condition without harming the image quality. The two examples provided in Figure 1b also demonstrate the flexibility of our method in customizing and combining multiple semantics. Furthermore, we show that our approach does *not* rely on any paired data or text annotations for training, but can still achieve fine-grained text-conditioned synthesis in the inference phase.

## 2 RELATED WORK

**Text-to-Image Synthesis.** The diffusion model (Ho et al., 2020; Song et al., 2020; Dhariwal & Nichol, 2021) has garnered considerable attention for its potential in image generation, particularly in the field of text-to-image synthesis. Previous works, such as GLIDE (Nichol et al., 2021) and DALLE2 (Ramesh et al., 2022b), have harnessed the power of CLIP (Radford et al., 2021) as an encoder to produce images with high text-matching accuracy. However, these models are limited in their ability to generate only low-resolution (64x64) images, necessitating additional super-resolution diffusion models (Saharia et al., 2022) for high-quality image generation. To address this limitation, latent-diffusion (Rombach et al., 2022) has been proposed, which involves pretraining an autoencoder and training a diffusion model in the latent space. These models have generated high-quality images using limited computational resources. Nevertheless, due to the constraints of CLIP, they tend to disregard details in the input text. Imagen (Ramesh et al., 2022a) uses a large, general-purpose language model (Raffel et al., 2020) to encode the text, improving the quality and consistency of text-to-image generation. Despite these improvements, current approaches still lack the ability to generate images with the desired level of detail. Our approach provides a significant improvement in generating high-quality images that closely align with the input text while also offering users more control over the level of detail in the generated images.

**Conditional Diffusion Model.** A diffusion model (Ho et al., 2020) consists of two processes: the forward process and the reverse process. The forward process is an explicit Markov process that

adds Gaussian noise to the real data distribution $x_0 \sim q(x_0)$ in a step-by-step manner:

$$q(x_{t+1}|x_t) := \mathcal{N}(x_{t+1}; \sqrt{1 - \beta_t}x_t, \beta_t\mathcal{I}), \quad t \in [0, T]. \tag{1}$$

Here, $\beta_1, \ldots, \beta_T$ are fixed constants, and $T$ is the number of time steps that is as large as possible to destroy the signal. The reverse process aims to approximate the posterior $p_\theta(x_{t-1}|x_t) := \mathcal{N}(x_{t-1}; \mu_\theta(x_t), \Sigma_\theta(x_t))$ by training a model $\epsilon_\theta$ with the training objective $|\epsilon_\theta - \epsilon|^2$. Some methods (Sohl-Dickstein et al., 2015; Vahdat et al., 2021; Dhariwal & Nichol, 2021) show that a pre-trained diffusion model can be guided by a classifier by adding its gradient to the mean value of images. GLIDE (Nichol et al., 2021) achieves text-conditioned image generation using a pre-trained noisy CLIP as the classifier:

$$\hat{\mu}_\theta(x_t|c) = \mu_\theta(x_t|c) + s\Sigma_\theta(x_t|c)\nabla(f(x_t) \cdot g(c)), \tag{2}$$

where $s$ is the guidance scale, $f(\cdot)$ and $g(\cdot)$ are CLIP image and text encoders respectively. GLIDE also uses classifier-free guidance (Ho & Salimans, 2022) by replacing the condition $c$ with a null condition randomly during training, and the reverse process is impelled close to $\mu_\theta(x_t|c)$ and away from $\mu_\theta(x_t|\emptyset)$ by:

$$\hat{\mu}_\theta(x_t|c) = \mu_\theta(x_t|c) + s \cdot (\mu_\theta(x_t|c) - \mu_\theta(x_t|\emptyset)). \tag{3}$$

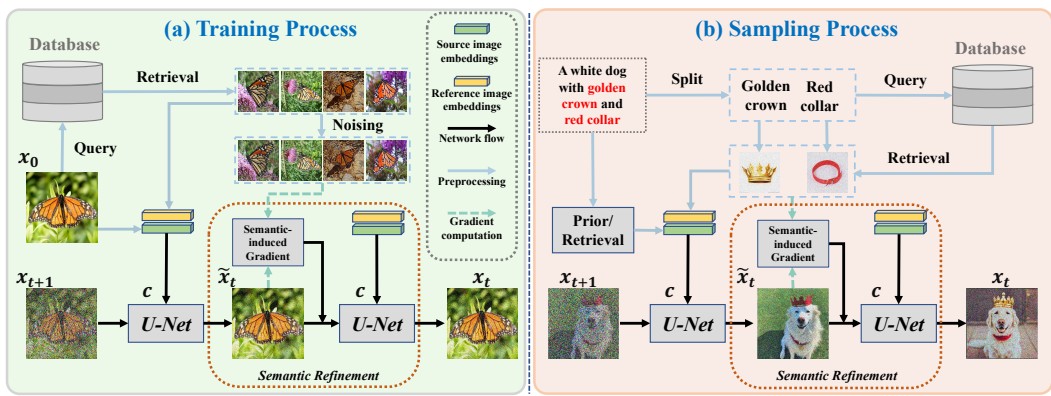

Figure 3: **Overview** of our proposed *SeReDiff*, which is based on diffusion models and adopts a *language-free training strategy* together with a *two-stage denoising process*. Given a source image $x_0$, we first retrieve some reference images from the database, which are used to help the model better understand the semantics in $x_0$. The CLIP embeddings from $x_0$ and the retrieved images are used as the condition of the first-stage denoising to generate a coarse image $\widetilde{x}_t$. Then, we compute semantic-induced gradient (see Section 3.3) from $\widetilde{x}_t$ and the noised version of reference images, and employ such gradient as an addition input for the second-stage denoising, *i.e.*, serving as *semantic refinement*. At the inference stage, we replace the image-based retrieval with text-image score matching *w.r.t.* the sementics of interests.

# 3 METHOD

## 3.1 OVERALL FRAMEWORK

**Training Process.** Our method aims to generate a high-quality image that accurately captures the highlighted details from the input sentence. To achieve this, we propose a novel diffusion-based generation framework, named *SeReDiff*. In a typical diffusion denoising process, the current image $x_t$ is directly predicted from the next image $x_{t+1}$ using a *U-Net* with a given condition $c$. To improve this process, as shown in Figure 3 (a), we expand the input space of the *U-Net* from 3 channels to 6 channels and split the denoising process into two stages. In the first stage, a null image of the same size as $x_{t+1}$ is concatenated with the image from the previous time step $x_{t+1}$ to predict a coarse image $\widetilde{x}_t = f_\theta(x_{t+1}, \emptyset, t, c)$, where $f_\theta(\cdot)$ denotes the *U-Net*, whose weights are shared in the two stages. In the second stage, the coarse image $\widetilde{x}_t$ is concatenated with the gradient to predict the denoised image $x_t$, *i.e.*, $x_t = f_\theta(\widetilde{x}_t, g, t, c)$. Here, $g$ denotes the *semantic-induced gradient*, which is computed as described in Sec 3.3.

**Sampling Process.** During the sampling process, as shown in Figure 3 (b), we aim to generate an image that aligns well with a given text description, while also incorporating the highlighted details. Firstly, we use the entire sentence to obtain a source image embedding by either using a prior or retrieving it from a database. Next, we employ the highlighted details to retrieve reference images, and concatenate the embeddings of the source image and reference images. By using concise terms to search for reference images, we ensure a strong alignment with the textual description. These concatenated embeddings serve as the conditions for the *U-Net* to synthesize the coarse image $\widetilde{x_t}$. In the second stage, we compute the semantic-induced gradient using a noised version of the reference images and the coarse image $\widetilde{x_t}$. We then use this gradient, in conjunction with the coarse image, to generate the image $x_t$. This process continues from time step $T$ until the final image is synthesized, with all semantics employed to guide the sampling simultaneously.

## 3.2 SEMANTIC REFINEMENT WITH GRADIENT CONDITION

**Why using gradient?** Recall that our objective is to guide the diffusion sampling process toward specific semantic directions. One intuitive approach is to directly concatenate the words extracted from the input sentence into the cross-attention module of the *U-Net* and use a classifier for guidance to generate the composite image. However, this approach has limitations in producing images with all fine-grained details, as different conditions have varying degrees of semantic relevance, and the more salient ones may dominate over the less prominent ones. Since the gradient has the same size as the generated image and implies a change in desired semantic direction, we can leverage it as a condition to guide the generation of the composite image. Previous works (Selvaraju et al., 2017; Fu et al., 2020) have shown that fine-grained details can be obtained from the gradient of a classifier. Therefore, we explore the potential of using the gradient as a condition to expand the 3-channel image space into a 6-channel gradient and image space in the diffusion model, with the goal of generating images with fine-grained details while maintaining semantic coherence.

**How does gradient facilitate semantic refinement?** Given a text prompt $c$ as condition, the conditional diffusion process (Dhariwal & Nichol, 2021) can be expressed as:

$$P_{\theta,\varphi}(x_t|x_{t+1}, c) = Z P_\theta(x_t|x_{t+1}) \cdot P_\varphi(c|x_t). \tag{4}$$

Here, $Z$ is a constant, $P_\theta(x_t|x_{t+1})$ represents the original diffusion process as a Gaussian distribution, $P_\varphi(c|x_t)$ is the posterior distribution for the condition $c$, which we will analyze in detail in the following.

Assuming the classifier posterior distribution $P_\varphi(c|x_t)$ is smooth with respect to variable $x_t$, this highly-nonlinear term can thus be further decomposed into a more tractable form: the combination of a linear term and a quadratic term. This decomposition serves as our key insight in designing *Semantic Refinement* with gradient condition. By applying Taylor expansion at $x_t = \mu$, we can get

$$\log P_\varphi(c|x_t) = C_0 + (x_t - \mu)\nabla_{x_t} \log P_\varphi(c|x_t)|_{x_t=u} + \frac{1}{2}(x_t - \mu)^T \nabla_{x_t}{}^2 \log P_\varphi(c|x_t)|_{x_t=\epsilon}(x_t - \mu), \tag{5}$$

where both $C_0$ and $\log P_\varphi(c|x_t)|_{x_t=u}$ are constant, $\epsilon \in [\mu, x_t]$ is a variable, $\nabla_{x_t}$ and $\nabla_{x_t}^2$ denote the gradient and Hessian operator *w.r.t.* variable $x_t$, respectively. For simplicity, in the following we denote $\nabla x_t \log P_\varphi(c|x_t)|_{x_t=u}$ as $g$ and $\nabla x_t{}^2 \log P_\varphi(c|x_t)|_{x_t=\epsilon}$ as $G(x_t, g)$.

Thus, Equation 4 can be represented as

$$\log(P_\theta(x_t|x_{t+1}) \cdot P_\varphi(c|x_t)) = \log p(z) + \frac{1}{2}g^T \Psi g + C_1, z \sim \mathcal{N}(\mu + \Psi g, \Psi^{-1}), \tag{6}$$

where $\Psi$ denotes $(\Sigma^{-1} - G(x_t, g))^{-1}$ and $C_1$ is a constant. Please refer to appendix A.1 for a detailed derivation. A key insight of Equation 6 is that, for a given $x_t$, the term $\frac{1}{2}g^T \Psi g$ is constant at each timestep. Thus, the entire conditional transition distribution can be approximated as a Gaussian distribution, where the variance is denoted as $\Psi^{-1}$ and the mean is represented as:

$$\hat{\mu}(x_t|c) = \mu_\theta(x_t|c) + \Psi g, \tag{7}$$

This formulation represents an advancement over classical results (Sohl-Dickstein et al., 2015; Dhariwal & Nichol, 2021), which only considered the first linear term to capture the conditional transition. Whereas our method incorporates the second order term elegantly due to the Gaussian nature of the mathematics involved and can capture the conditional transition more efficiently.

Given that $\Psi g$ is related to both $x_t$ and $g$, the mean value in Equation 7 can be modeled as a neural network:

$$\hat{\mu}(x_t|c) = \phi_\theta(x_{t-1}, t, g, c) = \mu_\theta(x_t|c) + \psi_\theta(g), \tag{8}$$

where $\mu_\theta(x_t|c)$ predicts the mean of the unconditional process and $\psi_\theta(g)$ is used to capture $\Psi g$.

**Property of using gradient.** Benefiting from the intrinsic linearity of the gradient operator, our method presents a valid compositional performance. Given multiple semantic $c_1, c_2, ..., c_n$ as conditions (Hinton, 2002), the conditional probability can then be factored as

$$P_{\theta,\varphi}(x_t|x_{t+1}, c_1, c_2, ..., c_n) = ZP_\theta(x_t|x_{t+1}) \cdot P_\varphi(c_1, c_2, ..., c_n|x_t) = ZP_\theta(x_t|x_{t+1}) \cdot \prod_{i=1}^{n} P_\varphi(c_i|x_t). \tag{9}$$

Here, $g_i$ denotes the gradient of the $i^{th}$ condition. Using Equation 6, we can compute the conditional probability of multiple conditions as $\mathcal{N}(\mu + \Psi \sum_{i=1}^{n} g_i, \Psi^{-1})$. This approach allows for the efficient and accurate implementation of the conditional diffusion process, facilitating fine-grained generation under multiple semantics.

### 3.3 SEMANTIC-INDUCED GRADIENT

To leverage gradient as a guiding signal in the diffusion sampling process, it is necessary to compute it efficiently and effectively. Previous work on this topic, such as GLIDE (Nichol et al., 2021), computes the dot product of noised image features and corresponding text features. However, this approach requires pre-training of noised CLIPs for images at each time step, which is computationally expensive and time-consuming. Instead, we use a loss function of negative dot similarity between CLIP embeddings of a generated image $\widetilde{x}_t$ and reference images $x_t^{ref}$. The gradient of this loss w.r.t. the generated image embedding is computed to obtain the semantic-induced gradient:

$$g = \nabla_{\widetilde{x}_t} f(\widetilde{x}_t) \cdot f(x_t^{ref}), \tag{10}$$

where $f(\cdot)$ denotes the CLIP image encoder. This way, we can efficiently guide the diffusion process towards specific semantic directions without pre-training noisy CLIPs

To ensure that CLIP can work well with noised images, we introduce a metric called the *Preservation Ratio of Representation Topology*. This metric measures the extent to which image feature similarity is maintained as the time step increases during the diffusion process. Specifically, for an image $S_0$, we calculate its CLIP similarity between $A_0$ and $B_0$. If $f(S_0) \cdot f(A_0) > f(S_0) \cdot f(B_0)$ and this condition holds at time step $t$ such that $f(S_t) \cdot f(A_t) > f(S_t) \cdot f(B_t)$, we consider $A_0$ at time step $t$ as "preserved" (and "not preserved" otherwise). We evaluate the Preservation Topology Ratio on a random sample of 1000 images from FFHQ and ImageNet datasets. The statistical results in Figure 4 demonstrate that even at time step 1000, the Preservation Topology Ratio for ImageNet is still about 80%, and over 82% for FFHQ dataset. Moreover, we observed that the

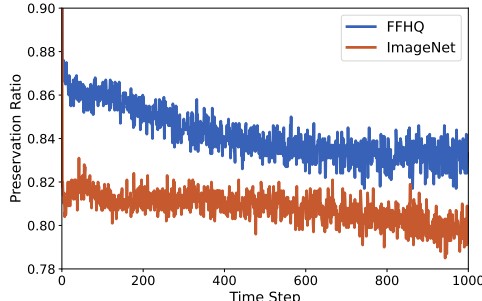

Figure 4: Analysis on preservation ratio of representation topology, which is obtained by computing CLIP feature similarity extracted from 1000 image pairs from each dataset. We compute their similarity by gradually destroying each image pair via adding noise, the same as the diffusion process in diffusion models.

similarity scores for images considered as "not preserved" are close to each other even at time step 0. This finding supports the feasibility of computing the semantic-induced gradient directly from two noised images, as an alternative to calculating it from image-text pairs.

**Language-free training.** The proposed semantic-induced gradient makes our training free of text annotations and allows for a language-free training pipeline where only pure images participate in the training process. Specifically, given an initial image $x_0$, we first extract its CLIP embedding $e$ with CLIP image encoder and retrieve its K-nearest neighbors $e^1, e^2, ..., e^k$ from the training database. Then we concatenate all the CLIP embeddings to form the condition

$c = \texttt{concat}(e, e^1, e^2, ..., e^k)$. The retrieved images corresponding to these embeddings serve as reference images $x^1, x^2, ..., x^k$. We use the noised versions of reference images $x_t^1, x_t^2, ..., x_t^k$ to compute the gradients $g_1, g_2, ..., g_k$ with Equation 10. Finally, we add the gradients to get the final gradient $g$ according to Equation 9.

## 4 EXPERIMENTS

### 4.1 EXPERIMENTAL SETUPS

**Datasets.** In this section, we compare our method with state-of-the-art approaches (Zhou et al., 2022; Xia et al., 2021a; Xu et al., 2018; Nichol et al., 2021; Liu et al., 2022) on various datasets containing fine-grained text descriptions, including Mulit-Modal CelebA-HQ (MM HQ) (Xia et al., 2021b) and CUB (Wah et al., 2011). For open-world results, we train our model on LAION-5b dataset (Schuhmann et al., 2022). To demonstrate the independence of our method from text-image pairs, we also present qualitative results obtained by training on AFHQ (Choi et al., 2020) and LHQ (Skorokhodov et al., 2021).

**Evaluation metrics.** To evaluate the visual quality of our proposed method, we use the Fréchet Inception Distance (FID) (Heusel et al., 2017) as a quantitative metric. However, since FID may not always align with human perception and does not necessarily reflect the semantic consistency between the image and text, we additionally conduct a user study to assess the photorealism, text alignment, and detail matching of the generated images. Specifically, users are given a text description that includes marked attributes and the corresponding generated image, and they are asked to rate the photorealism and text alignment of the image and calculate the number of matching attributes between the text and image. We evaluate our approach against several state-of-the-art methods, including TediGAN (Xia et al., 2021a), LAFITE (Zhou et al., 2022), AttnGAN (Xu et al., 2018), GLIDE (Nichol et al., 2021), Composable Diffusion (Liu et al., 2022) and Stable Diffusion (Rombach et al., 2022). Additionally, we use two baselines for ablation: (1) one-stage diffusion model without semantic refinement ($Baseline$) with identical parameter settings and training steps, and (2) $Baseline$ model with classifier guidance ($Baseline^C$). For further details regarding implementation and evaluation settings, please refer to Appendix A.2 and A.3.

### 4.2 QUANTITATIVE RESULTS

We quantitatively compare our proposed method with state-of-the-art benchmarks on MM-HQ (Lee et al., 2020) CUB (Wah et al., 2011) and Laion-5b (Schuhmann et al., 2022) datasets for fine-grained text-to-image generation. For the MM-HQ dataset, we compare our method with StyleGAN-based (Karras et al., 2019) methods LAFITE (Zhou et al., 2022) and TediGAN (Xia et al., 2021a). As shown in Table 1, our method outperforms the StyleGAN-based methods in terms of image quality, achieving the lowest FID score of 37.81 among all compared methods. Additionally, our method achieves the highest performance in Text Alignment and Detail Matching, with a matching score of over 90%, indicating that most fine-grained details described in the corresponding sentence are generated. However, our method is slightly inferior to TediGAN (Xia et al., 2021a) in terms of photorealism. The main reason is that our method only generates images with a resolution of $256 \times 256$ compared to TediGAN's $1024 \times 1024$ resolution. Note that analytical results have shown that users prefer higher resolution images. For CUB dataset, we compare our method with AttnGAN (Xu et al., 2018) and LAFITE (Zhou et al., 2022), and we find that our method outperforms them in all evaluation metrics.

### 4.3 QUALITATIVE RESULTS

**Comparison with State-of-The-Art methods.** Figure 5 displays the visual results of face images generated by previous methods. LAFITE (Zhou et al., 2022) can produce coarse-grained attributes such as "gender" and "hair style", but fails to capture fine-grained attributes like "sad", "green eyes", and "bushy beard". The quality of the generated images also degrades significantly when irregular attributes like "red beard" are involved. In contrast, TediGAN (Xia et al., 2021a) can generate high-quality images, but the semantics of the generated images do not always match the corresponding text descriptions. Our method outperforms both LAFITE and TediGAN, as it can

Table 1: Quantitative comparison of different methods on MM-HQ (Lee et al., 2020) and CUB (Wah et al., 2011) Dataset. ↓ means smaller number is better, while ↑ means larger number is better.

| Dateset | MM-HQ (Lee et al., 2020) | | | | CUB (Wah et al., 2011) | | | |
|---|---|---|---|---|---|---|---|---|
| Method | FID↓ | Photo realism↑ | Text Alignment↑ | Detail Matching↑ | FID↓ | Photo realism↑ | Text Alignment↑ | Detail Matching↑ |
| TediGAN | 40.48 | **4.23** | 4.15 | 77.48% | - | - | - | - |
| AttnGAN | - | - | - | - | 23.98 | 1.65 | 4.02 | 71.30% |
| LAFITE | 39.06 | 3.26 | 4.54 | 89.00% | 31.15 | 1.50 | 3.66 | 63.05% |
| Baseline | 39.74 | 3.99 | 4.49 | 87.83% | 22.03 | 1.50 | 3.27 | 50.35% |
| Baseline$^C$ | 39.76 | 3.79 | 4.46 | 86.32% | 21.62 | 1.73 | 3.19 | 48.85% |
| **Ours** | **37.81** | 4.00 | **4.59** | **90.45%** | **17.66** | **4.32** | **4.10** | **74.10%** |

synthesize high-quality images with fine-grained details that accurately reflect the meaning of the given text descriptions. Additionally, our method generates more realistic and natural results, such as "square face", "hat", and "red beard". Figure 6 showcases the visual results of bird images, where our method stands out in terms of photo-realism and text alignment compared to other methods.

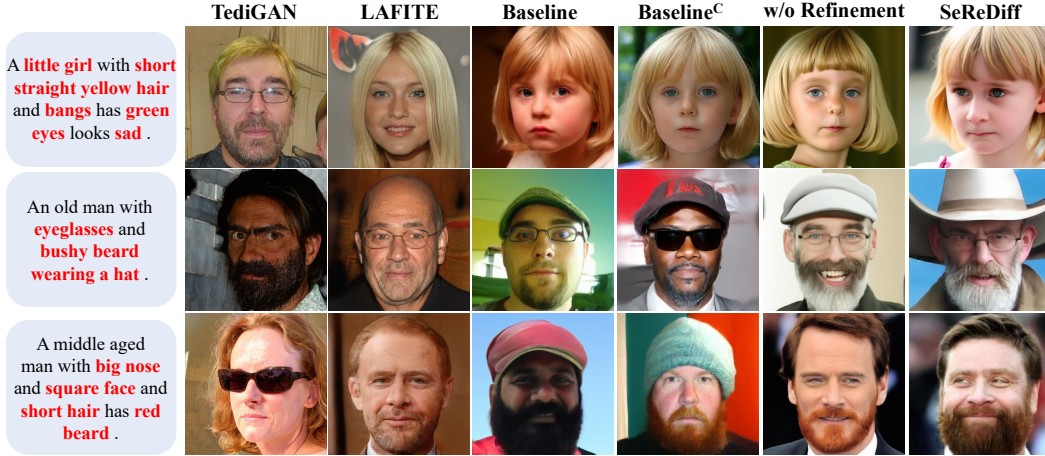

Figure 5: Qualitative comparisons on MM-HQ dataset (Lee et al., 2020). This first colume shows the input text prompt as the condition, while the remaining columns are samples produced by various methods. The highlighted words indicated the semantics to be strengthened.

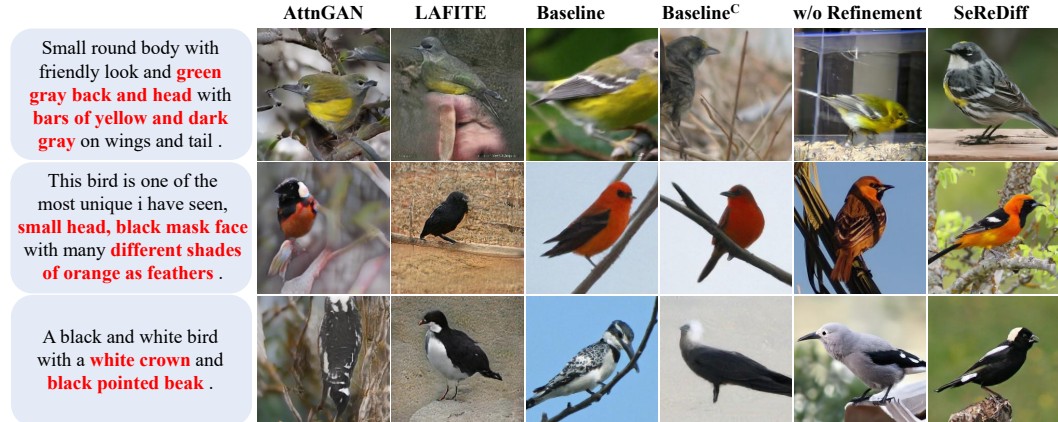

Figure 6: Qualitative comparisons on CUB dataset (Wah et al., 2011). This first colume shows the input text prompt as the condition, while the remaining columns are samples produced by various methods. The highlighted words indicated the semantics to be strengthened in our approach.

Table 2: Quantitative comparison on composable text-to-image generation. Compose+X denotes the implementation of composable diffusion using method X.

| Method ↓ | Photo Realism ↑ | Text Alignment ↑ | Detail Matching ↑ |
|---|---|---|---|
| Compose+GLIDE | 4.81 | 3.86 | 75.23% |
| Compose+stable diffusion | 4.81 | 3.89 | 75.52% |
| Structure diffusion | 4.78 | 3.81 | 71.06% |
| **Ours** | **4.85** | **4.62** | **91.69%** |

**Open-world Text-to-Image Generation.** We trained our model on LAION-5b (Schuhmann et al., 2022) dataset and pre-trained a prior for mapping CLIP text features into image features to facilitate open-world generation. To evaluate our approach, we compare it with the pre-trained models released by GLIDE (Nichol et al., 2021), Composable-Diffusion (Liu et al., 2022), and Stable Diffusion (Rombach et al., 2022). Figure 7 presents the visual results of the different methods. The results indicate that GLIDE (Nichol et al., 2021) is only capable of synthesizing a portion of the semantics, as demonstrated by the example in which the description is "A blue bird and a flower," and GLIDE only synthesized "a blue bird" while neglecting "a flower". Similarly, Composable-Diffusion (Liu et al., 2022) generates images with mixed-up semantics, such as a furry couch generated from the description "A couch and a dog sitting in the living room." While Stable Diffusion produces higher-quality visuals, it still fails to capture fine-grained details such as "golden crown" and "stars". Interestingly, our method successfully generates images that compose all the semantics, including counter-intuitive ones such as "A panda eats French fries with a red hat on the grassland."

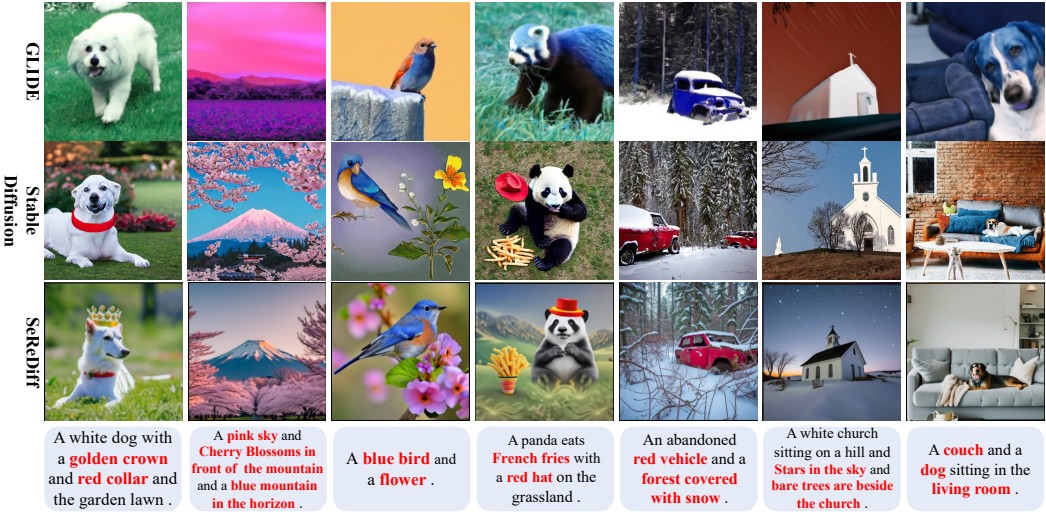

Figure 7: Illustrations of open-world text-to-image synthesis. The last row represents the input texts with semantics of interest highlighted, while the rest rows shows the results of different methods.

### 4.4 COMPOSABLE TEXT-TO-IMAGE GENERATION.

Our method supports composable text-to-image generation, which allows users to combine multiple textual descriptions into a single image. To evaluate the performance of our method in synthesizing fine-grained details, we compare it with two state-of-the-art composable text-to-image synthesis methods: Composable diffusion (Liu et al., 2022) and Structure-diffusion (Feng et al., 2022). Since different training sets are employed by these methods, we cannot directly compare the FID scores with them. Therefore, we mainly rely on human evaluations, which are presented in Table 2. Our method outperforms the others in Photo Realism, Text Alignment and Detail Matching, achieving a 91.69% rate for Detail Matching, which is a 16.46% improvement over the best existing method.

The qualitative results of different methods are shown in Figure 8. All of the prompts are selected from the paper of Structure-diffusion. The results of Composable diffusion (Liu et al., 2022) show low quality and entangled concepts. For example, in the prompt "A yellow cat and wearing a blue

plastic", our method clearly separates the cat and blue plastic, while Composable-diffusion mixes them together. Our method generates comparable results to Structure-diffusion (Feng et al., 2022) in image quality but shows better disentanglement of concepts. For instance, in the prompt "A gold clock and a green bench", Structure-diffusion produces a gold-green clock.

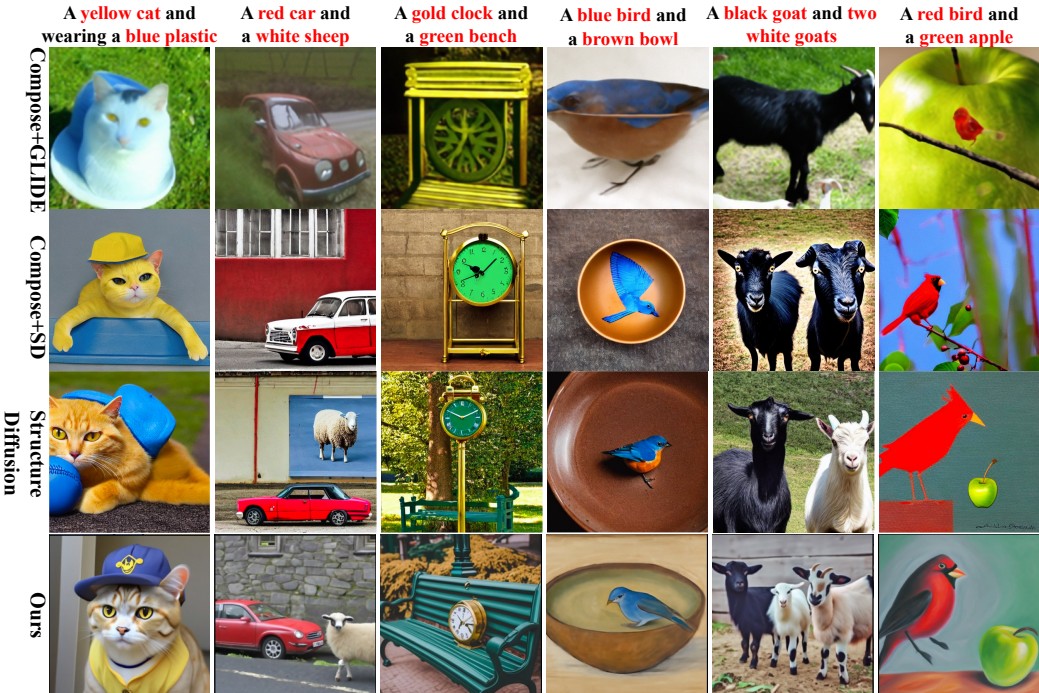

Figure 8: Illustration of different methods on composable text-to-image generation. Compose+X denotes the implementation of composable diffusion using method X.

### 4.5 ABLATION STUDIES

**Quantitative Evaluates.** As presented in Table 1, our approach outperforms the $baseline$ across multiple metrics, including Image Quality, Text Alignment, and Detail Matching. Specifically, our method achieves significant improvements in Image Quality, Text Alignment, and Detail Matching, particularly on the CUB dataset, where Photorealism increases from $1.50$ to $4.10$, and the Detail Matching is improved by $23.75\%$. While using classifier guidance ($Baseline^C$) only slightly improves the image quality. The poor results in Text Alignment and Detail Matching demonstrate the limitations of this method.

**Qualitative Evaluates.** The qualitative results of our method and ablations are presented in Figure 5 and Figure 6. The $Baseline$ method neglects fine-grained details, such as "square face" and "red beard" in face images, and produces overall blurry results. Although $Baseline^C$ performs better in modeling image details with sharper textures, the image distribution is severely damaged. In contrast, our method without Semantic Refinement produces results that are visually plausible and compatible with the conditioned texts. By incorporating the *Semantic Refinement* technique, we can obtain more fine-grained results, as shown in the rightmost column of the figures. Additionally, the semantics of the generated samples are enhanced, such as "hat," and "red beard" in Figure 5 and "white crown" and "black mask face" in Figure 6.

## 5 CONCLUSION AND DISCUSSION

In this paper, we propose a new method called SeReDiff that addresses the limitations of existing text-to-image methods in generating fine-grained images. Our approach uses a coarse-to-fine framework, generating a coarse-grained image first and refining it with a Semantic Refinement module that leverages multiple semantics. Our method offers language-free training and outperforms state-of-the-art methods on multiple datasets. It also allows users to emphasize desired semantics for flexible and customizable image synthesis.

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

# A APPENDIX

## A.1 DETAILED FORMULATION OF SEMANTIC REFINEMENT

In this section, we present a detailed formulation of semantic refinement. We begin by expressing the conditional diffusion transition as follows:

$$P_{\theta,\varphi}(x_t|x_{t+1}, c) = Z P_\theta(x_t|x_{t+1}) \cdot P_\varphi(c|x_t). \tag{a}$$

Here, $P_\theta(x_t|x_{t+1})$ is a Gaussian distribution with mean $\mu$ and covariance matrix $\Sigma$, given by:

$$P_\theta(x_t|x_{t+1}) = \mathcal{N}(\mu, \Sigma), \tag{b}$$

$$\log(P_\theta(x_t|x_{t+1})) = -\frac{1}{2}(x_t - \mu)^T \Sigma^{-1}(x_t - \mu). \tag{c}$$

Moreover, the classifier posterior distribution $P_\varphi(c|x_t)$ can be expanded around $x_t = \mu$ using a Taylor series expansion as follows:

$$\log P_\varphi(c|x_t) = C_0 + (x_t - \mu)\nabla_{x_t} \log P_\varphi(c|x_t)|_{x_t=u} + \frac{1}{2}(x_t - \mu)^T \nabla_{x_t}^2 \log P_\varphi(c|x_t)|_{x_t=\epsilon}(x_t - \mu), \tag{d}$$

To simplify the notation, we use $g$ and $G(x_t, g)$ to denote $\nabla_{x_t} \log P_\varphi(c|x_t)|_{x_t=u}$ and $\nabla_{x_t}^2 \log P_\varphi(c|x_t)|_{x_t=\epsilon}$, respectively.

$$
\begin{aligned}
&\log(P_\theta(x_t|x_{t+1}) \cdot P_\varphi(c|x_t)) \\
&= -\frac{1}{2}(x_t - \mu)^T \Sigma^{-1}(x_t - \mu) + (x_t - \mu)g + \frac{1}{2}(x_t - \mu)^T G(x_t, g)(x_t - \mu) + C_1 \\
&= -\frac{1}{2}(x_t - \mu)^T (\Sigma^{-1} - G(x_t, g))(x_t - \mu) + (x_t - \mu)g + C_1 \\
&= -\frac{1}{2}(x_t - \mu - \Psi g)^T \Psi^{-1}(x_t - \mu - \Psi g) + \frac{1}{2}g^T \Psi g + C_1 \\
&= \log p(z) + \frac{1}{2}g^T \Psi g + C_1, \quad z \sim \mathcal{N}(\mu + \Psi g, \Psi^{-1}).
\end{aligned} \tag{e}
$$

In this equation, $\Psi = (\Sigma^{-1} - G(x_t, g))$ and $C_1$ is a constant.

## A.2 IMPLEMENTATION DETAILS

In our experimental setup, we employ 8 A100 GPUs to conduct experiments on various datasets such as FFHQ (Karras et al., 2019), CUB (Wah et al., 2011), AFHQ (Karras et al., 2019), and LHQ (Skorokhodov et al., 2021). Additionally, we use 48 A100 GPUs to perform experiments on the Laion-5b (Schuhmann et al., 2022) dataset. To extract features, we utilize a ViT-L/14@336px (Radford et al., 2021) CLIP image encoder and text encoder.

For training the diffusion model, we adopt the U-Net architecture used in the Dalle2 (Ramesh et al., 2022b) decoder. The input and output dimensions of the U-Net are set to 6 and 3, respectively, but

we scale the model width to 196 channels. The attention layer consists of 64 head channels with an attention resolution of 8, 16, and 32. We set the learning rate to $1.2e-4$. During training, we randomly set the condition and gradient to $\emptyset$ with probabilities of 10% and 50%, respectively. We set the total diffusion timesteps to 1000. We also train a prior on Laion-5b dataset with parameters introduced in (Ramesh et al., 2022b). During the sampling stage, we adopted the DDIM sampling strategy with a sampling step of 50. Additionally, we incorporated classifier guidance with a guidance scale of 3.

### A.3 HUMAN EVALUATION DETAILS

In our user study, we conduct a human evaluation to assess the performance of different image generation methods on the CUB and MM-HQ datasets. We use three evaluation metrics: Photorealism, Text Alignment, and Detail Matching. Fifty individuals participated in the evaluation process.

To evaluate Photorealism and Text Alignment, users are presented with a sample image and its corresponding text description. They are then asked to rate the Photorealism and Text Alignment of the image on a scale of 1 to 5. To simplify the evaluation process, we divide Photorealism into three levels: Completely broken image (1 point), Image with artifact or blur (3 points), and Image without artifact and blur (5 points). Similarly, for Text Alignment, we divide the evaluation into four levels: Complete mismatch (1 point), Mismatch with most of the semantics (2 points), Match with most of the semantics (4 points), and Complete match (5 points). For Detail Matching, we mark out the details in the text description, and users are asked to indicate the number of details satisfied in the sample image. The percentage of matched details is then calculated. Finally, we calculate the average rating for each method based on all user ratings, which served as the final score.

### A.4 LANGUAGE-FREE GENERATION.

Our approach leverages a language-free training strategy, enabling it to be trained on datasets without text annotations, including AFHQ (Choi et al., 2020) and LHQ (Skorokhodov et al., 2021) datasets. The upper row of Figure 9 displays some LHQ-generated samples. As observed in the odd columns, our method without semantic refinement can generate the general semantics of the input text descriptions. However, fine-grained details such as "pool," "mountains," and "streams" are missing. In contrast, the proposed method with semantic refinement, shown in the even columns, largely replenish these missing details. In the lower row of Figure 9, we demonstrate samples generated from the AFHQ dataset. Our method captures the color of hair, ear shapes, and eye and mouth states accurately.

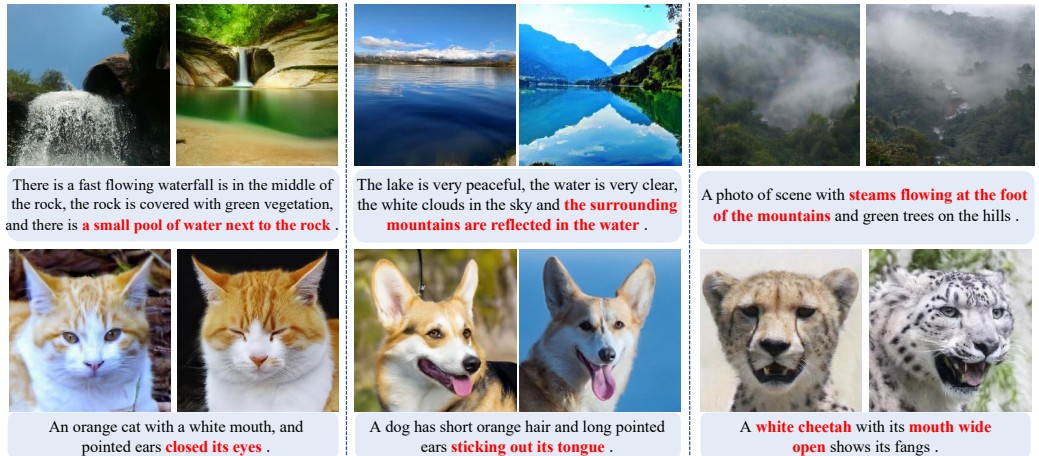

Figure 9: Language-free training results of SeReDiff trained on LHQ (upper row) and AFHQ (lower row). The odd columns show the images conditioned on the entire sentence, while the even columns present the results produced after semantic refinement.

## A.5 EFFECTIVENESS OF SEMANTIC REFINEMENT.

We demonstrate the effectiveness of the proposed Semantic Refinement mechanism in enhancing fine-grained details by progressively incorporating semantics, as shown in Figure 10. As more semantics are incorporated, the generated images become increasingly refined and better aligned with the given text description. Additionally, fine-grained details, such as "pointed nose" and "beard" are emphasized.

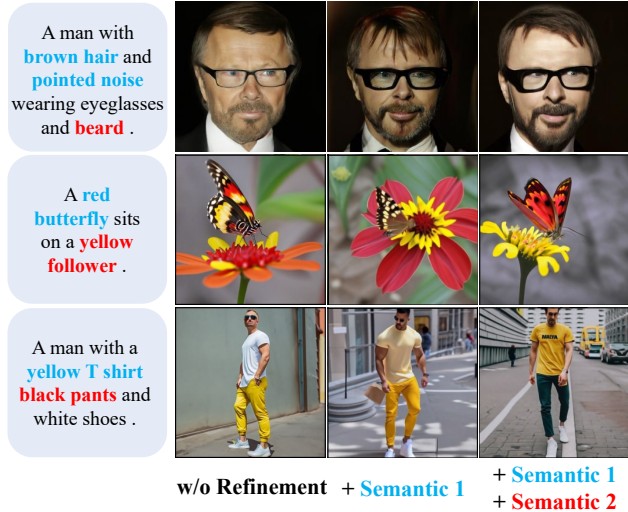

Figure 10: Illustration of the refinement process by emphasizing varying number of semantics.

## A.6 DIVERSE RESULTS.

To demonstrate the capability of our method in generating diverse outputs based on a given text description, we conducted further experiments on several datasets, including Laion-5b, LHQ, FFHQ, and AFHQ. The results of our qualitative analysis are presented in Figure 11, Figure 12, and Figure 13.

*A robot ballerina dancing in a flower field at night with the moon in the background*

*a white dog and a kid with blue T shirt and hat are sitting on the lawn.*

*At sunset, abandoned boats sit on the beach and flocks of birds fly across the sky*

*The Totoro bus is driving on the water*

*An yellow dog with eagel's wings is flying over the sea*

*An airship shaped like a pig floating over a wheat field, the tractor on the ground*

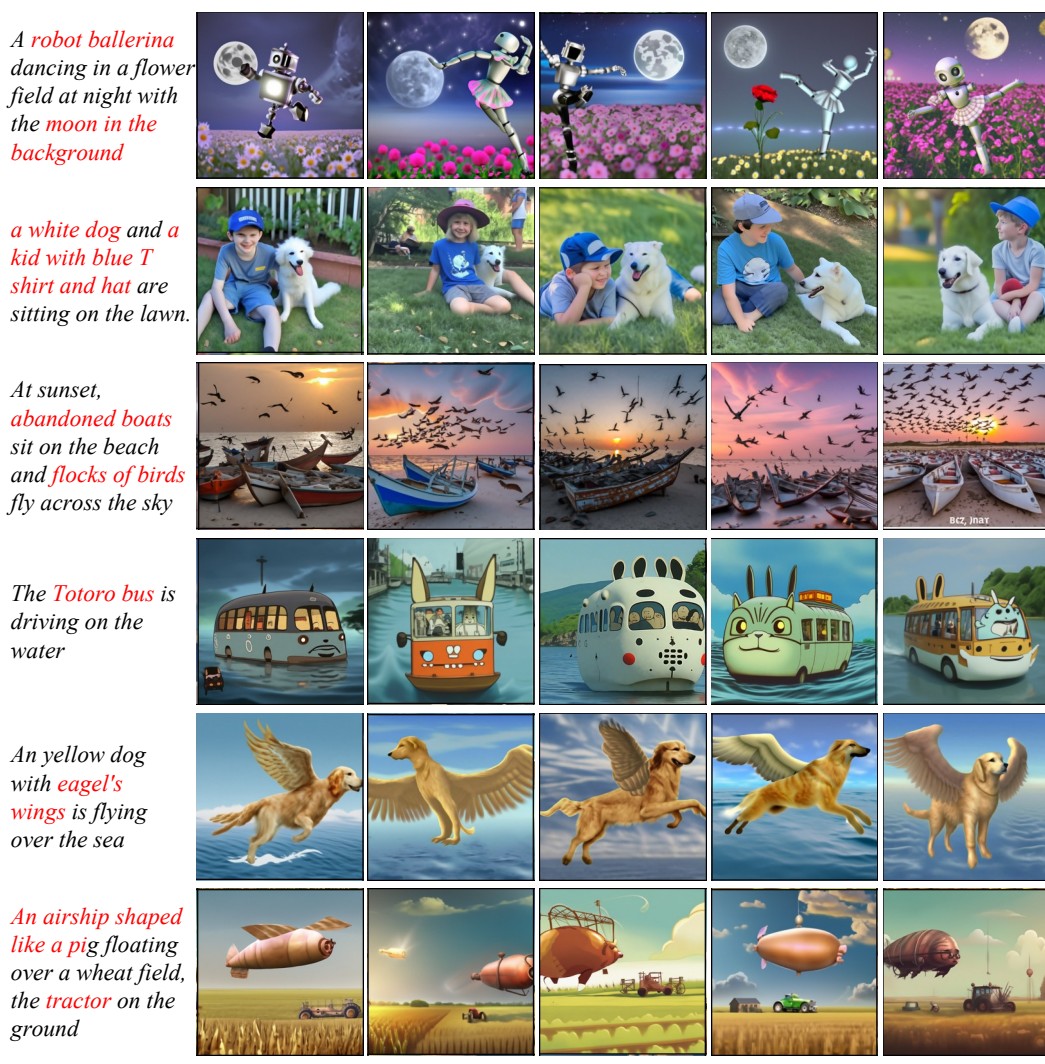

Figure 11: Illustrations of diverse open-world results. The left column represents the text descriptions, and the rest of column are generated results with given text.

*a scenic picture with white clouds in the sky and dense forests reflected in the water is a delight to the eyes*

*a picture of snow covered peaks towering into the clouds*

*a picture of a sunset with snow falling on the trees*

*On the calm sea, there is a towering rock, and the sunset in the distance reflects the beautiful light*

*Under the blue sky is a white snowy mountain, and the reflection of the snow capped mountain is reflected on the calm lake*

*mist shrouds the forest, the sun casts a little shimmer, and only a few leaves float on the trunks of the trees*

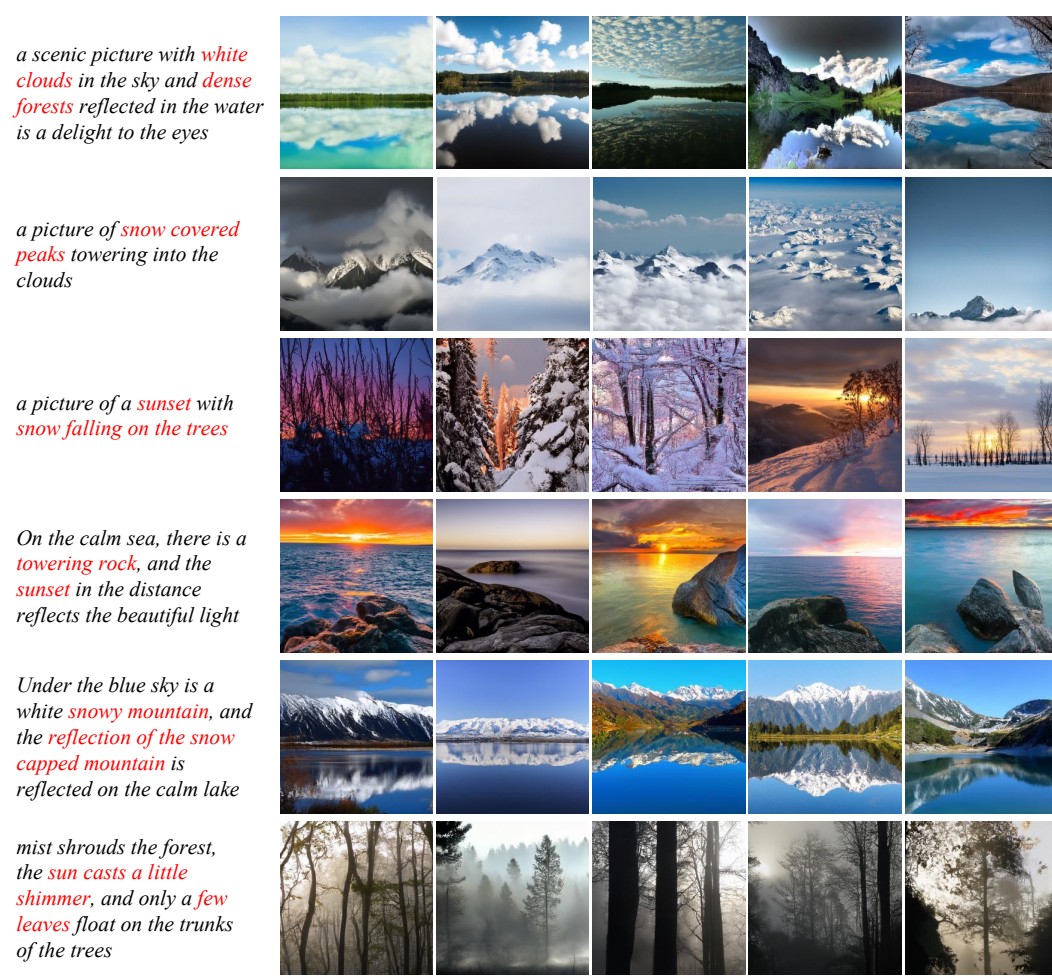

Figure 12: Illustrations of diverse results trained on LHQ dataset. The left column represents the text descriptions, and the rest of column are generated results with given text.

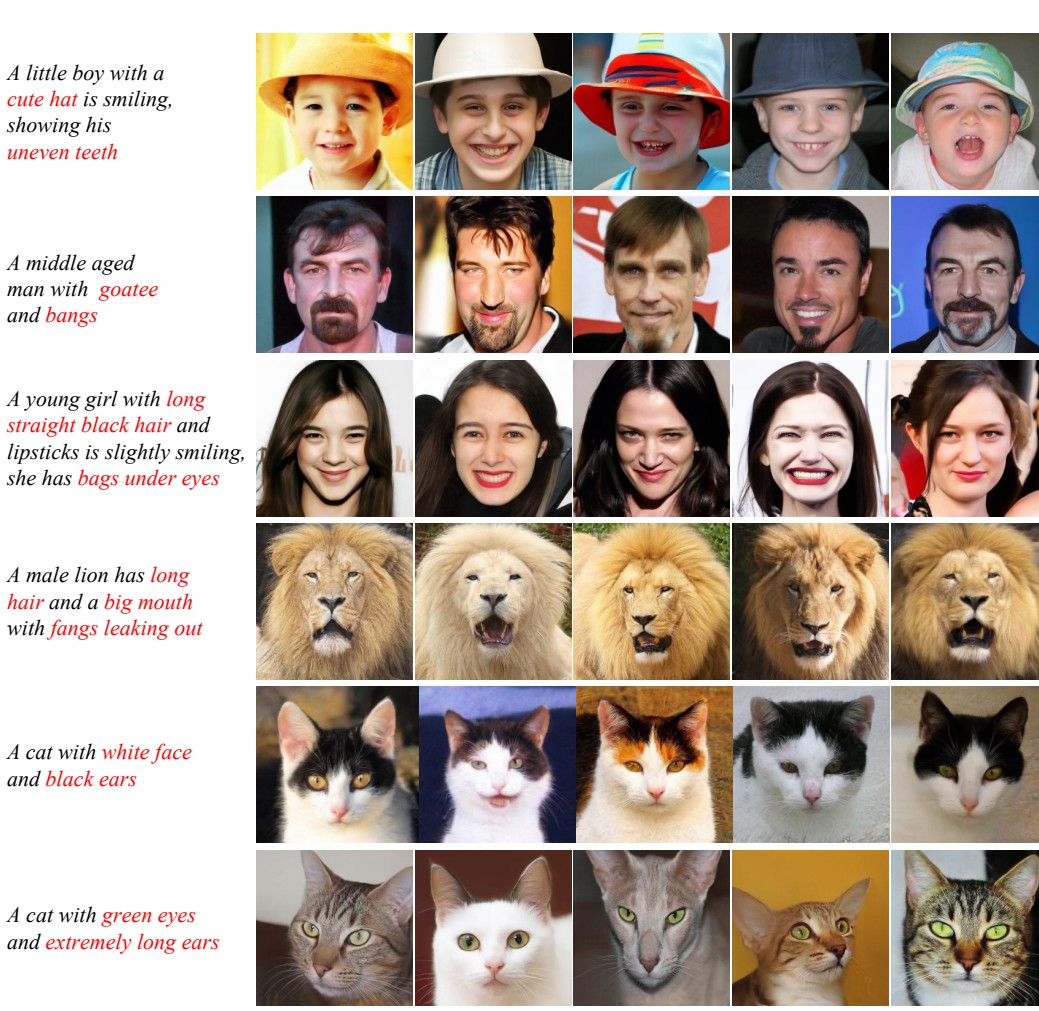

*A little boy with a cute hat is smiling, showing his uneven teeth*

*A middle aged man with goatee and bangs*

*A young girl with long straight black hair and lipsticks is slightly smiling, she has bags under eyes*

*A male lion has long hair and a big mouth with fangs leaking out*

*A cat with white face and black ears*

*A cat with green eyes and extremely long ears*

Figure 13: Illustrations of diverse results trained on FFHQ and AFHQdataset. The left column represents the text descriptions, and the rest of column are generated results with given text.

