# OpenReview forum: "Fine-grained Text-to-Image Synthesis with Semantic Refinement"
_ICLR.cc/2024/Conference — Submitted to ICLR 2024_

### Official Review · Reviewer_AM6U · 2023-10-22

**Soundness:** 3 good
**Presentation:** 3 good
**Contribution:** 2 fair
**Rating:** 5
**Confidence:** 4

**Summary:**

This study introduces a diffusion-based technique that allows for fine-grained synthesis with semantic refinement. Rather than synthesizing from an entire descriptive sentence, users can highlight specific words. A semantic-induced gradient is incorporated at every denoising phase to assist the model in understanding the given words.

**Strengths:**

The paper is well-written and easy to understand.

The innovative approach of emphasizing specific words to enhance text-to-image synthesis is novel.

The inclusion of a user study offers valuable subjective evaluations.

**Weaknesses:**

A primary concern is the limited comparison with recent methods. Considering the rapid advancements in the text-to-image research field (from GAN to Transformer to Diffusion in just three years), a sufficient comparison becomes more important. I commend authors  for leveraging the large-scale LAION dataset, which could reflect the image generation capabilities well. Could authors further include comparisons on this dataset with models like Make-A-Scene, DALL-E 2, CogView 2, Imagen, Parti, or Re-Imagen?

There's a notable absence of the FID metric on LAION dataset. Even with potential variations across training datasets, there remains a requirement for an objective quantitative evaluation of all models. I believe this metric should be made accessible to the readers. In addition, incorporating the CLIP score, if possible, would be a commendable step.

The paper utilizes an additional database for reference image retrieval, enhancing its SEMANTIC REFINEMENT capability. Therefore, it's crucial to provide insights into how this database was constructed. It would also be beneficial to address how the methodology performs with unusual semantics (e.g., Pikachu) or distinct styles (e.g., Picasso-esque).

The novelty of the paper seems somewhat limited. The method primarily relys on CLIP for condition guidance. This dependence might influence the model's performance, especially in distinguishing subtle differences between similar images.

**Questions:**

During the testing phase, how does this work automatically emphasize specific words within the sentence?

---

### Official Review · Reviewer_qacC · 2023-10-30

**Soundness:** 2 fair
**Presentation:** 2 fair
**Contribution:** 2 fair
**Rating:** 3
**Confidence:** 4

**Summary:**

**Summary:**
The paper introduces a retrieval-based method for text-to-image generation. The authors argue that this approach can better align semantics.

**Strengths:**

**Advantages:**
1. The writing quality of the paper is acceptable, although the foundational formulation doesn't introduce anything particularly novel.
2. While the language is clear, the paper doesn't sufficiently distinguish its method from prior similar approaches.

**Weaknesses:**

**Disadvantages:**
1. The most significant issue with this paper is its lack of innovation.
2. The results presented are just average. The methods compared are those published more than a year ago, and recent methods are notably absent.

**Questions:**

**Concerns:**
1. There have already been numerous methods based on retrieval. The core idea of this paper closely mirrors that of "Re-Imagen". It's challenging to discern any unique innovation. The authors' arguments on contributions don't point to any significant novelties. Moreover, the paper does not discuss differences from methods like "Re-Imagen", emphasizing its lack of substantial innovation.


**References:**
@article{chen2022re,
  title={Re-Imagen: Retrieval-Augmented Text-to-Image Generator},
  author={Chen, Wenhu and Hu, Hexiang and Saharia, Chitwan and Cohen, William W},
  journal={arXiv e-prints},
  pages={arXiv--2209},
  year={2022}
}

@inproceedings{koh2021text,
  title={Text-to-image generation grounded by fine-grained user attention},
  author={Koh, Jing Yu and Baldridge, Jason and Lee, Honglak and Yang, Yinfei},
  booktitle={Proceedings of the IEEE/CVF winter conference on applications of computer vision},
  pages={237--246},
  year={2021}
}

@article{sheynin2022knn,
  title={Knn-diffusion: Image generation via large-scale retrieval},
  author={Sheynin, Shelly and Ashual, Oron and Polyak, Adam and Singer, Uriel and Gafni, Oran and Nachmani, Eliya and Taigman, Yaniv},
  journal={arXiv preprint arXiv:2204.02849},
  year={2022}
}

@article{blattmann2022semi,
  title={Semi-parametric neural image synthesis},
  author={Blattmann, Andreas and Rombach, Robin and Oktay, Kaan and M{\"u}ller, Jonas and Ommer, Bj{\"o}rn},
  journal={Advances in Neural Information Processing Systems},
  volume={11},
  year={2022}
}

@inproceedings{liu2023more,
  title={More control for free! image synthesis with semantic diffusion guidance},
  author={Liu, Xihui and Park, Dong Huk and Azadi, Samaneh and Zhang, Gong and Chopikyan, Arman and Hu, Yuxiao and Shi, Humphrey and Rohrbach, Anna and Darrell, Trevor},
  booktitle={Proceedings of the IEEE/CVF Winter Conference on Applications of Computer Vision},
  pages={289--299},
  year={2023}
}

---

### Official Review · Reviewer_mHH3 · 2023-10-31

**Soundness:** 3 good
**Presentation:** 3 good
**Contribution:** 2 fair
**Rating:** 5
**Confidence:** 5

**Summary:**

This paper aims to address the problem of misalignment between the text prompt and the generated image in current t2i models. The authors propose to incorporate a semantic-induced gradient into the denoising process to facilitate the alignment of text details and synthesized images.  Extensive experiments on several datasets verify the effectiveness of the proposed method.

**Strengths:**

1. The detailed analysis of why the text-to-image generation model fails to correctly match the text when it is complex is well done. （Figure 2)
2. The visualizations in Figure 5&6&7  demonstrate good text-image alignment, indicating the effectiveness of the proposed method.

**Weaknesses:**

1. From the quantitative results in Table 1, we can see that the improvements in text alignment and detail matching are not impressive compared with other methods. For MM-HQ dataset, the proposed method performs similarly to LAFITE on both text alignment (the proposed method: 4.59 vs LAFITE : 4.54) and detail matching (the proposed method: 90.45% vs LAFITE : 89%) metrics. For CUB dataset, the proposed method also performs similarly to AttnGAN. Therefore I have some concerns about the effectiveness of the proposed method despite its qualitative results seeming good.
2. The idea of using text prompts to retrieve similar images from a database as a condition for image synthesis bears some similarities with "Retrieval-Augmented Diffusion Models" and "KNN-Diffusion: Image Generation via Large-Scale Retrieval". Although the motivation of the proposed method seems different, this similarity diminishes the originality of the paper.
3. There are no quantitative results for open-world text-to-image synthesis. It's preferable to randomly sample a validation dataset to assess the effectiveness of the proposed method across a wider spectrum.

**Questions:**

1. The qualitative results (last row) in Figure 7 indicate that the generated images look blurrier than those from stable diffusion. Could the authors give a detailed analysis?
2. The proposed method needs a database to get the reference image, therefore the diversity of the database is very important. What if there is no similar image that matches the sub-text prompt correctly?

---

### Official Review · Reviewer_KP2Z · 2023-10-31

**Soundness:** 3 good
**Presentation:** 2 fair
**Contribution:** 3 good
**Rating:** 6
**Confidence:** 3

**Summary:**

The paper introduces SeReDiff, a novel diffusion-based method that overcomes the limitations of existing text-to-image methods in generating fine-grained images that closely match the input text condition, especially lengthy text. SeReDiff also empowers users to emphasize some specific words to guide the generation more accurately. To achieve this purpose, the authors restructure the denoising pipeline of diffusion models by integrating the semantic-induced gradient as a reference input at each denoising step, alongside the image denoised from the previous step. Furthermore, the proposed approach does not rely on text annotations for training but can still achieve fine-grained text-conditioned synthesis in the inference phase.

**Strengths:**

_ The authors provide appropriate analyses of the CLIP model, including the imperfect text-image matching as well as the preservation ratio of representation topology.

_ The idea of using gradient to facilitate semantic refinement is interesting and it is complemented by a comprehensive mathematical proof.

_ Extensive experiments are conducted to showcase the robustness of SeReDiff in generating high-quality images that align well with the input text condition when compared to many existing methods.

**Weaknesses:**

_ The paper is quite difficult to follow.

_ The experiment displayed in Table 2 lacks information regarding the generating resolution of each model, whereas this detail can affect the quality of the output images.

**Questions:**

_ Have you considered applying your approach to the latent space instead of pixel space?

_ When incorporating both database retrieval and a two-stage denoising process, does the proposed method significantly extend its runtime in comparison to other methods?

---

### Meta-Review · Area_Chair_kMjV · 2023-12-06

**Metareview:**

The paper receives mixed reviews. The reviewers raise concerns about the novelty, limited experimental results, and missing comparisons to most recent methods. The authors did not provide a rebuttal. The authors are encouraged to address these issues and resubmit to another venue.

**Justification For Why Not Higher Score:**

The authors did not provide a rebuttal to address the issues raised by the reviewers.

**Justification For Why Not Lower Score:**

N/A

---

### Decision · Program_Chairs · 2024-01-16

Reject